# A Review on Application of Acoustic Emission in Coal—Analysis Based on CiteSpace Knowledge Network

Shankun Zhao [1,2], Qian Chao [2], Liu Yang [3] , Kai Qin [1] and Jianping Zuo [2,4,*]

1 Mine Safety Technology Branch of China Coal Research Institute, Beijing 100013, China
2 School of Mechanics and Civil Engineering, China University of Mining and Technology (Beijing), Beijing 100083, China
3 State Key Laboratory for Geomechanics and Deep Underground Engineering, China University of Mining and Technology (Beijing), Beijing 100083, China
4 State Key Laboratory of Coal Resources and Safe Mining, China University of Mining and Technology, Beijing 100083, China
* Correspondence: zjp@cumtb.edu.cn; Tel.: +86-189-1039-7078

**Abstract:** Based on CiteSpace software, this paper reviews and analyzes the application articles of acoustic emission in coal from 2010 to 2020. In this paper, CiteSpace software visualizes 453 articles collected in the Web of Science core database. The cooperation networks between different countries, institutions, and authors are used to determine the connection of knowledge in papers. The keyword co-occurrence, keyword co-occurrence time zone map, and keyword clustering are used to determine the hot topics in the field. The cited collaborative network analysis reveals the important literature and the contribution of prominent authors in this area. In the future, for the research of acoustic emission in coal mining, compression tests will still be the main test methods. In terms of time domain parameters of acoustic emission, the application of ring counting, energy, waveform, and signal strength are very mature. The principal problem of acoustic emission location operation will become a focus in the future. The most widely used patterns in the determination of ruptures are the signal intensity fractal dimension, the acoustic emission number, and the b-value. In practical engineering problems, there is little research on the deformation activity law of steeply inclined coal seams and surrounding rock. The mining of steeply inclined coal seams is still a difficult problem. There are immature technologies in coal mining, rockburst early warning, and coal and gas outburst. In terms of the intellectualization and accuracy based on experience, there is room for improvement in the future. Scholars will continue a deeper exploration on the application of the numerical simulation.

**Keywords:** acoustic emission; coal; CiteSpace; knowledge network; future research; visual analysis

## 1. Introduction

When the coal breaks under the action of external and internal force, the strain energy is released in the form of an ultrasonic wave, resulting in the ultrasonic wave phenomenon. This is the essence of acoustic emission. When coal produces the acoustic emission phenomenon, it will be accompanied by deformation and fracture, and the application of acoustic emission technology is very important at this time. Acoustic emission technology employs real-time continuous monitoring of microscopic damage generation and expansion in materials. It has been widely used in many projects, such as tunnel, slope, hydropower, mine ground pressure, and safety monitoring. In the stability analysis of mine tunnels, it is very important to predict the dynamic disaster of coal and monitor the evolution process of the internal damage of coal.

In terms of experiments, scholars have conducted much experimental research on the problem of dynamic disasters in coal mining by using acoustic emission technology. Acoustic emission technology is combined with uniaxial compression, triaxial compression, permeability experiments, splitting experiments, three-point bending experiments, and

3D simulation. This paper expounds the changes in coal stability from the perspectives of energy, fracture evolution, and strength change. For example, some scholars have analyzed the influencing factors of on-site coal and gas outbursts through acoustic emission technology, and have studied the application of acoustic emission techniques in evaluating coal subsurface gasification models and operational parameters [1,2]. They described the gasification process using acoustic emission techniques, adopted more accurate prediction methods for prediction accuracy, studied the multi index discrimination model of coal and gas outbursts based on FDA, and applied the discrimination model to the prediction of coal and gas outbursts. The FDA method has broad application prospects in the prediction of coal and gas outbursts. In the study, the application of AE in evaluating the UCG model and operating parameters was discussed, and the gasification process was described according to the analysis results. The final results showed that AE monitoring was helpful in maintaining a safe and efficient UCG process. In 2019, some scholars carried out a uniaxial compression acoustic emission test on the experimental samples. They analyzed the mechanical properties of the specimens by using the acoustic emission technique in real time, obtained the basic mechanics and acoustic emission change laws of the sample in the whole fracture process, and analyzed the spatial evolution and analysis laws of acoustic emission [3]. The results show the relationship between the acoustic emission and the stress of the two samples.

Due to the complexity of deep geological conditions in practical engineering application, it is easy to cause structural disturbances in the surrounding rock during coal mining. It will cause roof fracture, fault sliding rockburst, and other problems. Therefore, it is particularly important to understand the rock characteristics in advance. For example, A. M. Naji and H Rehman et al. [4] simulated the dynamic phenomena of rockburst near the shear zone of the headrace tunnel by using the FLAC 2D explicit numerical program in 2019, and studied the behavior of rock mass around the tunnel under static and dynamic loads. Additionally, according to the modeling results, the load conditions of the support system in the adjacent tunnel affected by the dynamic rockburst were obtained. Finally, one of the most important rockburst control factors was clarified through numerical analysis, and yield support measures that can withstand the dynamic impact of rockbursts in deep hard rock tunnels were recommended.

The behavior of the ground is affected by the rock mass characteristics with geostress and groundwater, the cross section of the excavation area, the excavation method, and the excavation speed. These factors should be considered in order to ensure the ground support and stability during underground excavation. According to the Neelum Jhelum Hydropower Station Project, Rehman et al. [5] discussed the main challenges encountered during the construction and their countermeasures, and adopted the latest technology and methods of rock mechanics to complete the project. The excavation method used in the project plays a very important role in the ground composition and serious rockburst, and will help in the future complex tunnel engineering.

Some scholars used a combination of triaxial compression test and acoustic emission technique to explore fracture evolution process of argillaceous limestone samples under different initial confining pressures [6]. They studied the problem of fractured rock mass often encountered in underground buildings and how to restore the strength of fractured rock mass under uniaxial compression. After uniaxial compression, the rock mass is recovered by adhesive, and then the sample recovered by uniaxial compression is crushed. Using the cooperation between acoustic emission and the uniaxial compression system, the crack threshold of the sample is defined. The final research results provide guidance for the restoration of the internal mechanism in the process of rock fragmentation. Some scholars summarized the fracture evolution process and the change in permeability under coal loading by using the acoustic emission technique [7]. They conducted methane permeability tests of coal samples under different loading and unloading paths, quantitatively studied the influence of the unloading rate on the mechanical behavior and permeability evolution of coal, and obtained the rules concerning unloading rate and elastic energy, dissipated

energy, and permeability of coal. Li et al. studied the characteristics of acoustic emission signals in the process of hard roof failure, and found that the periodic characteristics and evolution process of acoustic emission signals can reflect not only the stress state of roof strata, but also the degree of roof failure, which provides a theoretical basis for practical engineering [8].

### 1.1. Monitoring Coal and Gas Outburst

Coal and gas outbursts often lead to major accidents in coal mines. Many researchers at home and abroad have studied the triggering mechanism of coal and gas outbursts through similar simulation tests, which provide theoretical support for prediction and early warning of coal and gas outbursts. Acoustic emission technology is often used to monitor coal and gas outbursts. In engineering, acoustic emission technology is often called microseism. Scholars have also conducted relevant research on coal and gas outburst. They often use microseismic monitoring data to confirm some mechanism problems of indoor acoustic emission experiments, and use acoustic emission experiments to study the waveform problems of microseisms. For example, in Zhu Quanjie et al., the energy distribution difference between the typical blasting vibration waveform and the rock fracture waveform in the multi-frequency band was studied by using wavelet packet analysis technology, which provided a new method and idea for the identification of microseismic signals [9]. They quantitatively analyzed the law of coal microseismic activity, which provides a strong theoretical basis for disaster prediction. Su Chengdong et al. conducted three different compression tests on coal samples and studied the difference by using acoustic emission numbers and energy [10]. Lei Wenjie et al. studied the triggering mechanism of coal and gas outbursts through a similar simulation test, and monitored the frequency signals of different outburst stages by using the high sensitivity of microseismic technology, providing a test basis for actual projects [11].

### 1.2. Reflect the Internal Crack Propagation of Rock

Acoustic emission technology is often used to reflect the crack extension of rock during loading, as well as the stress state. The combination of infrared radiation and acoustic emission technology can explore the mechanical properties of rocks during loading, and provide reference for field monitoring. Tan Jianuo et al. conducted uniaxial compression tests on the specimen [12]. They studied the crack evolution process in the rock by using the acoustic emission Ra value, and found its relationship with indoor rockbursts, which has a good guiding role for on-site monitoring and early warning of rockbursts. Han Xianmin et al. used the true triaxial test to simulate the actual scene of a rockburst [13]. They studied the threshold of the rockburst under the action of dynamic disturbance, summarized the law, and provided a favorable theoretical basis for practical engineering.

### 1.3. Basis of Water Inrush Monitoring and Prediction

Acoustic emission technology is also commonly used in mine water inrush early warning, and AE signal is often used as the basis of water inrush monitoring. Tang Shoufeng et al. used acoustic emission signal noise reduction processing, and selected the waveform with the best noise reduction effect [14]. They proposed signal feature extraction methods of the wavelet characteristic energy spectrum coefficient and the characteristic vector. These methods can study the change of rock mass stress in the process of water inrush. It lays an important foundation for analyzing the evolution process of water inrush acoustic emission events in coal mines from the level of the time series. Zhang Yanbo et al. used a high-speed camera, an infrared thermal imager, and an acoustic emission system to monitor the water inrush process of the roadway heading face, then compared the response sequence between the three [15]. Ultimately, it was proposed that due to the complexity of the actual situation, a joint monitoring approach with multiple devices is recommended for validation. This would provide more reliable early warning information for inrush water monitoring.

Some former scholars summarized and compared the early warning technology of coal and rockburst disasters according to big data from the occurrence mechanism [16]. Chen Yu et al. introduced the application of acoustic emission technology in rock mechanics from a simple principle to practical application, and analyzed the current achievements [17].

However, there is no systematic analysis focusing on specific areas of coal and acoustic emission. In this paper, CiteSpace is used to visually analyze the literature data of the application of acoustic emission in coal, so as to have a more comprehensive understanding of the research in this field by scholars at home and abroad. CiteSpace can make a quantitative analysis of the academic literature by using mathematical statistical methods, as well as scientific methods which are more accurate for co-occurrence or co-analysis. These remarkable features are widely used in various fields. CiteSpace is used to systematically analyze the literature related to acoustic emission and coal, according to the global publications from the core database of Web of Science from 2010 to 2020. It involves a series of angles, such as cooperation analysis, co-occurrence of keywords in the literature, and relevance for statistical analysis of the data, so as to determine the development process and future direction. This paper introduces the retrieval method and CiteSpace measurement software, then discusses the development process of acoustic emission in coal by using the scientific econometric analysis method. It also involves global publication output analysis, cooperation analysis, keyword analysis and clustering in China, and, finally, a summary of the important conclusions.

## 2. Literature Sources and Metering Methods

The research method of this paper is divided into three parts in order to retrieve the most authoritative and comprehensive relevant literature. In addition, CiteSpace is used to find mathematical statistics on the literature and thereby obtain the regular scientific map. Finally, this paper describes the future development research of acoustic emission technology in the field of coal mining. The following Figure 1 depicts the flow chart of this paper.

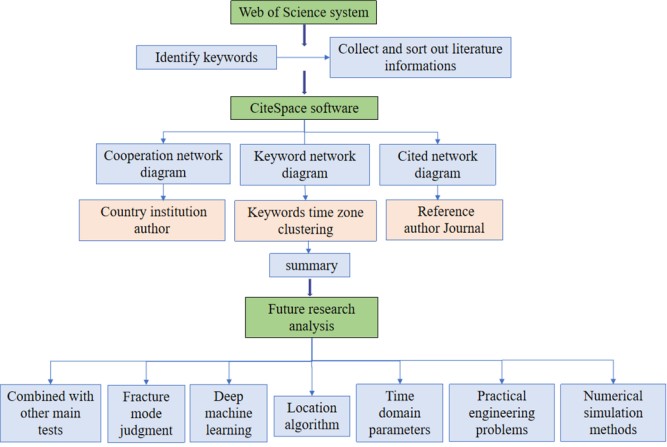

**Figure 1.** Research flow chart.

### 2.1. Source of Literature

Taking appropriate databases and keywords is the first step in carrying out the research. The literature collection uses the Web of Science core database as the literature source, which is an authoritative platform for national academic research and a large, comprehensive, multidisciplinary, and core journal database. The database is the most influential and high-quality database of more than 8000 of its kind around the world. It is very simple and comprehensive, using logic "and", "or", and "non" for retrieval. This paper adopts theme searches. The keywords are "acoustic emission" and "coal". The language type is "English". The literature type is "article and review", and the time span is nearly 10 years, accurate from 1 January 2010 to 31 December 2020. A total of 453 documents were obtained

by this method. The plain text files of "all records and references", "complete records", and Excel files of 453 documents were exported from the database for later data analysis.

### 2.2. Metering Method

This article uses CiteSpace version 5.8 R3 to carry out quantitative data analysis. CiteSpace can scientifically identify the content of documents and visually process research trends, hot spots, and other information. Weizhang Liang et al. [18] used CiteSpace software to conduct a scientific metrological review of hard rockburst research from 2000 to 2019. They systematically analyzed the research status of rockbursts and provided valuable and in-depth understanding in this field for researchers. Junzhi Zhang et al. [19] used CiteSpace software to scientifically measure a large number of documents on the service life of reinforced concrete structures, described the retrieval strategy of data collection, and carried out an information visualization analysis. In order to realize the risk assessment of mine water, Liu Beizhan et al. [20] used CiteSpace software to collate the relevant literature of the past 10 years, and analyzed the hot spots and frontier trends of mine water inrush research in China. They used the text files exported from the literature collection platform for scientific quantitative analysis. The colors shown in the figure also represent different meanings. As shown in Figure 2, the colors in the network diagram represent years. In addition, mathematical statistics and comprehensive scientific analysis are reflected in the figure according to the data. This includes "centrality", which indicates the importance of nodes. When its value is greater than 0.1, it plays a core role in the network diagram.

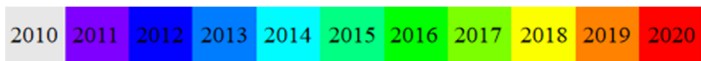

**Figure 2.** CiteSpace network chart color represents year.

### 2.2.1. Cooperative Networks (Country, Author, Institution)

The three cooperation networks were used to intuitively obtain the cooperation relationship between different countries, scholars, and institutions in this field. These cooperation networks can provide direction for scholars in collection of the literature and reflect the research value of this field.

### 2.2.2. Keyword Co-Occurrence, Keyword Co-Occurrence Network Time Zone Map, Keyword Clustering

The keywords of three visualization forms, taking the literature content as the background and the hot topics in this field as the key, more deeply describe the key and difficult problems in the research field. Keywords are the simplification of the retrieved articles. The frequency of keywords reflects the core content of the article, which plays a representative role in its selection.

### 2.2.3. Cited Collaboration Network (Article, Author, Journal)

The cited analysis step reveals the status of the research from three aspects: article, author, and journal. Through the cited collaborative analysis, we can come to learn the basic knowledge of this field. It is very helpful to obtain the important literature that plays a key role in this field, so as to obtain its development trend.

## 3. Quantitative Analysis Results of Data

According to the data analysis and scientific econometric analysis of network atlas, the specific research contents are visualized. The specific results are as follows:

### 3.1. Analysis of Document Quantity

#### 3.1.1. Output Analysis of Global Document Volume

It can be seen from Figure 3 that the output of the relevant literature on the application of acoustic emission in coal in the world is on the rise. In 2010, He, MC, and other

scholars studied the acoustic emission characteristics through the rockburst process of limestone under the condition of true triaxial unloading [21]. From 2010 to 2016, publications have changed slightly; obvious changes have taken place since 2016. It can also be seen from Table 1 that the total output of articles in the literature from 2010 to 2020 was 453; from 2016 to 2017, the number of documents increased from 21 to 52, and to 132 in 2020. Similarly, the number of citations to the literature increased greatly from 2016 to 2017, from 586 to 1425, a significant increase in the literature output. From 2017 to 2020, it has gradually entered the high-speed accumulation period of theoretical research results and practical exploration of acoustic emission in coal application. From the relative growth rate of the previous two years, it can be seen that the relative growth rate of publications has decreased, and the relative heat has also decreased in recent years from 2017 to 2020, though it still shows an increasing trend. The trend of the number of citations in Figure 3 and the increase in the published articles with the year indicate that the application of acoustic emission in coal is still a key research problem for scholars in various countries.

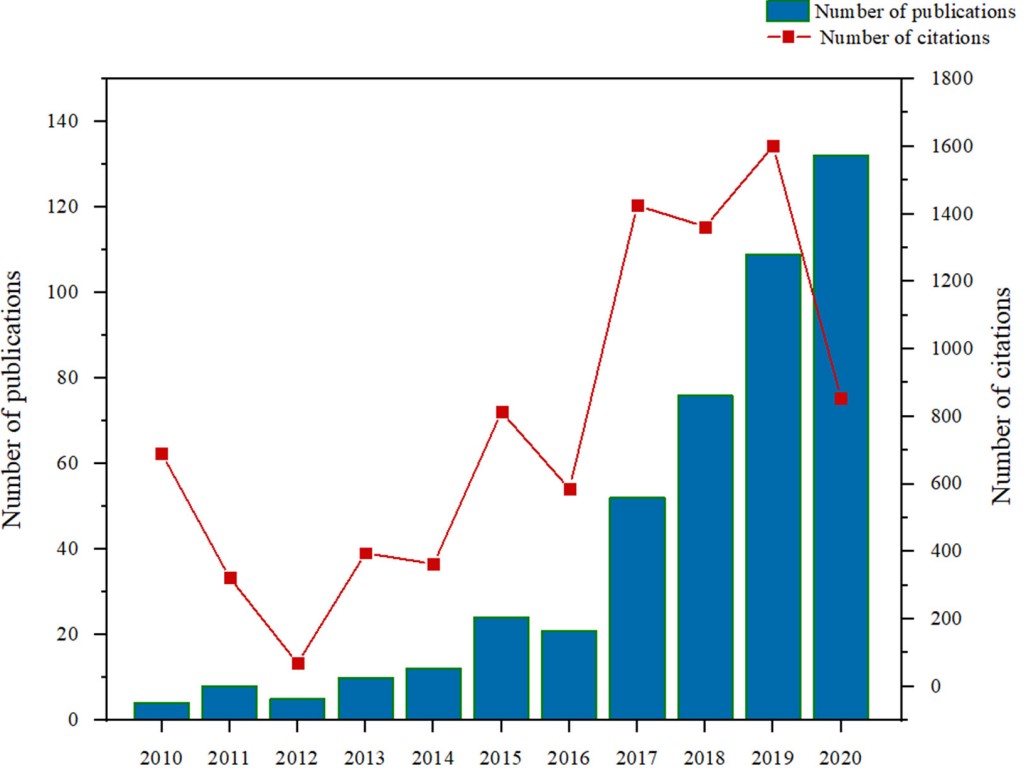

**Figure 3.** Number of publications and citations globally from 2010 to 2020.

**Table 1.** Analysis of global document volume from 2010 to 2020.

| Number | Year | Number of Global Publications | Relative Growth Rate | Cited Times |
|---|---|---|---|---|
| 1 | 2010 | 4 | / | 690 |
| 2 | 2011 | 8 | 100% | 322 |
| 3 | 2012 | 5 | −37.5% | 69 |
| 4 | 2013 | 10 | 100% | 396 |
| 5 | 2014 | 12 | 20% | 363 |
| 6 | 2015 | 24 | 100% | 814 |
| 7 | 2016 | 21 | −12.5% | 586 |
| 8 | 2017 | 52 | 148% | 1425 |
| 9 | 2018 | 76 | 46.2% | 1363 |
| 10 | 2019 | 109 | 43.4% | 1601 |
| 11 | 2020 | 132 | 21.1% | 854 |

### 3.1.2. Analysis on the Number of Articles Published by Chinese Authors

This part of the data is obtained by a web of science filtering countries. After visualization, it can be seen from Figure 4 that the change in the trend of the number of articles published by Chinese authors and the number of citations over the years is consistent with that of the world. The published articles generally show an upward trend. Similarly, great changes have taken place in the number of articles published and the number of citations in 2016. It can be found in Table 2 that from 2010 to 2020, the total number of articles published by Chinese authors was 408, accounting for 90% of the total number of 453 globally. Table 2 shows that the relative growth rate of publications from 2011 to 2012 is 0.00%, which is consistent with the change in the trend of national document volume in Table 1 over the past two years. This shows that China plays a very important role in this field. The reason for this phenomenon is that China is a large infrastructure country. Tunnels, slopes, hydropower, and other underground projects account are relatively abundant in the country. Dynamic disasters emerge one after another, such as mine ground pressure, the harm of coal and gas outbursts, rockbursts, and so on. In order to solve these problems, scholars attach great importance to the research in this field. For example, in 2021, Wang Wei et al. conducted a static load test on semi-circular (SCB) grooved coal samples. They studied the coupling effect of coal sample bedding and prefabricated crack direction on coal fracture mechanical properties [22]. The final results provide experimental support for the meso-mechanism research of some disasters in coal mining. At the same time, they can also provide technical guidance for engineering practice, such as reducing hydraulic fracturing costs, controlling the development of coal seam networks, and improving mining efficiency. Kong Xiangguo et al. carried out a triaxial compression test on coal containing formaldehyde. They studied the fractal characteristics and acoustic emission characteristics of coal damage evolution [23]. This research is helpful for understanding the influence of gas on coal and the evolution mechanism of cracks, and can be applied to the research on the occurrence mechanism of coal and gas outbursts. In the research of scholars, it is pointed out that the emission technique plays a very important role.

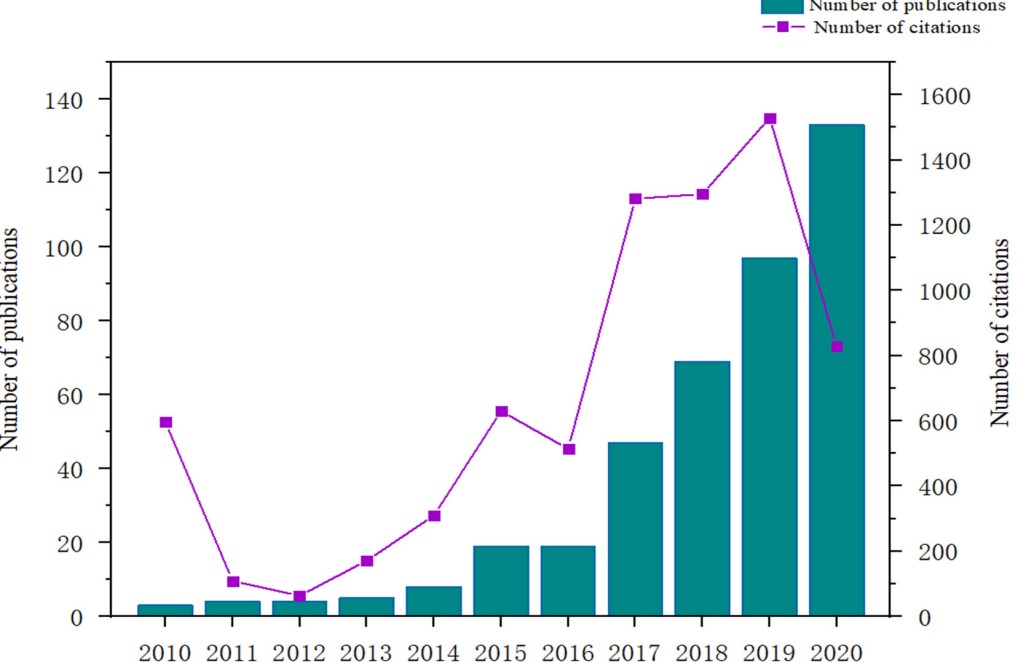

**Figure 4.** Number of publications and citations in China from 2010 to 2020.

**Table 2.** Analysis, including the number of articles published by Chinese authors from 2010 to 2020.

| Year | Including the Number of Articles Published by Chinese Authors | Relative Growth Rate | Cited Times |
|---|---|---|---|
| 2010 | 3 | / | 596 |
| 2011 | 4 | 33.3% | 109 |
| 2012 | 4 | 0.00% | 64 |
| 2013 | 5 | 25% | 171 |
| 2014 | 8 | 60% | 310 |
| 2015 | 19 | 137.5% | 630 |
| 2016 | 19 | 0.00% | 514 |
| 2017 | 47 | 147.4% | 1282 |
| 2018 | 69 | 46.8% | 1296 |
| 2019 | 97 | 40.6% | 1530 |
| 2020 | 133 | 37.1% | 829 |

*3.2. Cooperative Analysis*

The following will be analyzed from the three cooperation networks of countries, institutions, and authors. The nodes in the cooperation analysis diagram represent the amount of documents issued by a country, institution, or author. An article can contain multiple authors or countries; when two different authors or countries appear in the same article, a connection will appear between the two nodes in order to indicate that there is a cooperative relationship between the two. The color of the line corresponds to the color of the figure, and also represents the year. The thickness of the line represents the cooperation intensity [24]. The size of the nodes in the graph represents the number of documents of a certain author, country, or institution. The size of the nodes is directly proportional to the number of documents. The color of nodes in different cooperation network graphs represents their active years.

3.2.1. Country Cooperation Network

According to the analysis chart of the national cooperation network in Figure 5, China has the largest node, indicating that compared with other countries, the number of published documents is the highest. The top three countries are China, Australia, and the United States, and the betweenness centrality values are 0.72, 0.3, and 0.12, respectively. The data show that the number of papers published by Chinese, Australian, and American scholars are 398, 51, and 33, respectively. In the article by Zhang Li et al. [25] on the comparison between the situation regarding China and the global coal industry in 2021, it is shown that by the end of 2019, the global coal reserves were 1069636Mt. The article data showed that the top ten countries in the global coal resource reserves were the United States, Russia, Australia, China, India, European Union, Indonesia, Germany, Ukraine, and Poland (Figure 6). Among them, the top five countries with resource reserves were the United States (249537Mt, accounting for 23.33% of the world's total reserves), Russia (162166Mt, 15.16%), Australia (149079Mt, 13.94%), China (141595Mt, 13.24%), and India (105931Mt, 9.61%). The recoverable reserves of coal resources in the above five countries account for 75.57%, which is why the volume of publications is mainly concentrated in these countries.

Among these, 398 articles issued by China are inconsistent with the 408 articles in Table 2. After analysis, the reason is that the Web of Science system has nothing to do with the author order. For example, the author of a country ranks seventh in the article, which would still be included. However, CiteSpace software has corresponding restrictions in this regard, resulting in a corresponding reduction in the frequency of authors, but it can also explain the problem. In addition, from the color of the nodes in the Figure 5, it can be concluded that China first achieved research results in this field, and has close cooperative relations with Britain, Canada, the United States, Australia, and Japan.

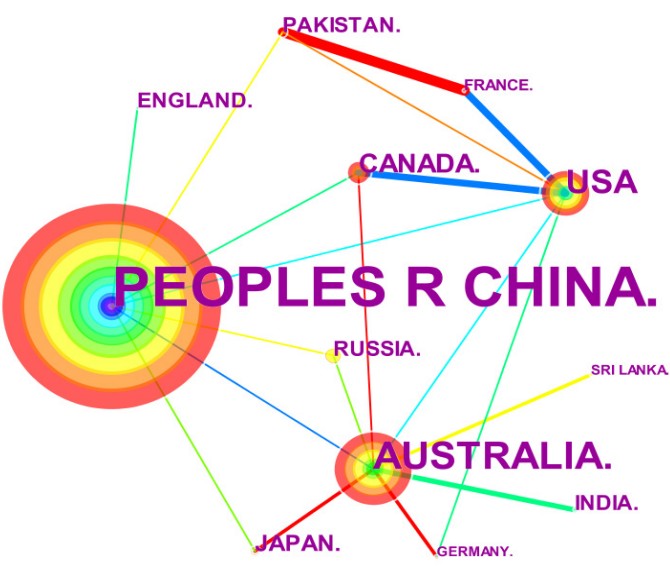

**Figure 5.** Country cooperation network.

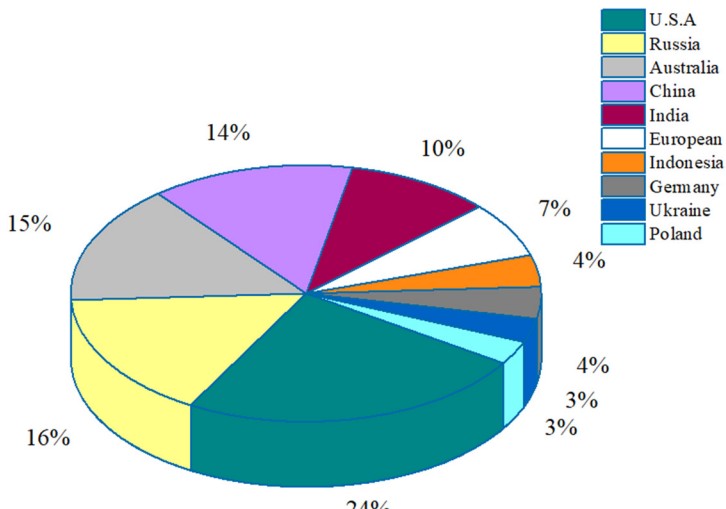

**Figure 6.** Proportion of the top 10 countries in global coal reserves in 2019.

### 3.2.2. Institution Cooperation Network

In the institution cooperation network (Figure 7) and the data, the top seven are China University of Mining and Technology, 168; Shandong University of Science and Technology, 44; Chongqing University, 42; Monash University, 28; Beijing University of Science and Technology, 27; Henan University of Technology, 24; and Xi'an University of Science and Technology, 18. China University of Mining and Technology is in a core position, with a centrality of 0.63. Mining and Engineering at China University of Mining and Technology is a national key discipline. It has very close cooperation with Central South University, Science and Technology University of Beijing, Xi'an University of Science and Technology, and other universities. It has two national key laboratories, one national engineering technology research center, and several key laboratories of the Ministry of Education. It has made very important contributions in the field of coal. For example, in the "12th Five-Year Plan" for scientific and technological development, it has studied the green mining technology of open-pit coal mines and developed domestic mobile crushers, electric wheel dump trucks, etc. In the manufacturing of sensing mine and coal mine equipment, the key technologies and equipment have been developed. In terms of clean coal and

transformation in utilization, underground coal gasification technology has been studied. Secondly, the centrality of Chongqing University is 0.23, ranking second. Figure 7 shows that mainly Chinese institutions are included, which corresponds to the data obtained from the national cooperation network in Figure 5. From this, it is concluded that all institutions have made great contributions to the application field of acoustic emission in coal.

**Figure 7.** Institution cooperation network.

### 3.2.3. Author Cooperation Network

The cooperation network diagram of the authors shows the cooperation relationship of acoustic emission in the application of coal in the most detail. From Figure 8, it is obvious that Professor Yuan Wang is the author with the largest number of papers, with 40 papers. Seven scholars, including Wang Yuan, Kong Xiangguo, and Li Zhonghui, have published more than 10 papers. A total of 25 scholars have five or more articles, which shows that there is a high concentration of authors in the application field of acoustic emission in coal. Combined with Figures 7 and 8, it is found that the institution of various scholars basically corresponds to the document-issuing institutions. For example, Professor Enyuan Wang teaches at the China University of Mining and Technology, mainly studying safety technology and engineering, as well as the prevention of mine gas and coal dynamic disasters (coal and gas outbursts, rockbursts). He jointly founded the electromagnetic radiation theory of coal damage and invented the acoustic and electrical monitoring for early warning technology of coal dynamic disasters. He put forward the mechanism of stress induced polarization (non-uniform variable-speed deformation of coal and rock) and electromagnetic radiation generated by the variable-speed movement of charged particles, invented the real-time monitoring and early warning technology of coal dynamic disasters, theoretically established the prediction criteria of coal and rock dynamic disasters, and invented the portable electromagnetic radiation monitor and the online electromagnetic radiation monitor for coal and gas outbursts. Professor Xiangguo Kong teaches at Xi'an University of Science and Technology, mainly studying coal and gas dynamic disasters, as well as geophysical signal response. He has strong cooperation with Professor Wang Enyuan of China University of Mining and Technology, and has published 37 articles in China's HowNet system, which has a great influence in this field.

### 3.3. Keywords Research and Analysis

Keyword co-occurrence takes the literature content as the background; the hot topics in this field as the key to more deeply describing the problems in the research field. This is performed through three methods: keyword co-occurrence network, keyword time zone network, and keyword clustering. For the first two, a node represents a keyword, and the size of the node represents the frequency of the keyword. The larger the node, the

more frequently the keywords appear, and they are in direct proportion. The color of the node represents the year of publication. When keywords appear in the same document, a connecting line will appear between them. The color of the line corresponds to the year, its thickness represents the co-occurrence intensity, and the color of the line also represents the time since the keywords appeared.

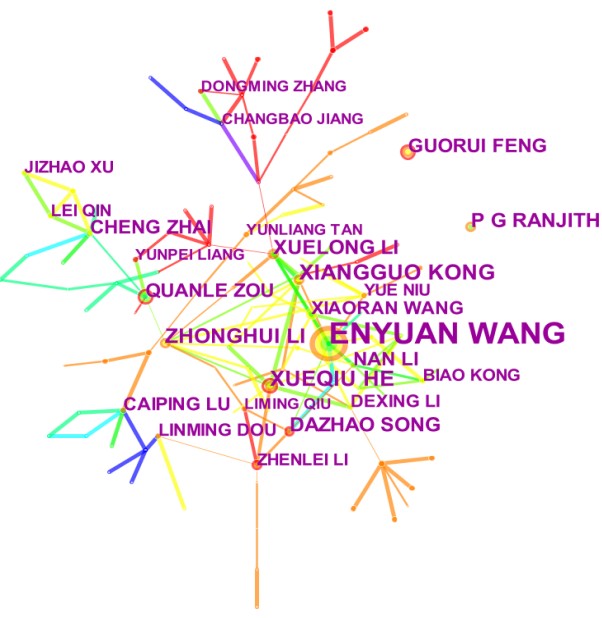

**Figure 8.** Author cooperation network.

### 3.3.1. Keywords Co-Occurrence Network

From the keyword co-occurrence network (Figure 9), it is obvious that "rock", "acoustic emission", "coal", and "behavior" are in the core position. The occurrence frequencies are relatively high, with values of 125, 121, 111, and 90 times, respectively, while the centrality values are 0.15, 0.11, 0.21, and 0.12, respectively. Obviously, this is very consistent with the theme of this paper, and the application of acoustic emission technology in coal is also very common. For example, Zhang, HY et al. [26] used acoustic emission to study the effect of stress damage on the permeability of Beishan granite in 2020. They measured the gas permeability of Beishan granite under hydrostatic pressure and a cyclic loading and unloading triaxial test. In the test, acoustic emission technology reflected the microcrack growth process in granite samples well, and quantified the cumulative damage. Zhang, YB, et al. [27] used acoustic emission to conduct experimental research on the acoustic emission energy and the frequency characteristics of tensile and shear cracks during a tunnel rockburst in 2020. The authors proved the feasibility of wavelet feature coding and obtained the feature coding of the time sequence and the state of the time sequence signal at the same time, laying a foundation for analysis of the evolution of water inrush acoustic emission coal from the level of time sequence. In 2019, Cheng, Guanwen, et al. studied the microseismic phenomenon during the activation of a mining brittle fracture in a coal mine in China [28]. In this study, scholars analyzed the stress state and AE (acoustic emission)/MS (microseism) activity in the activation process of the brittle fault, the activation stage of the brittle fault, and the relationship between AE/MS activity and parameters of the brittle fault. It is feasible to identify the buried brittle fault, then to determine its parameters and activation process based on MS monitoring. The final conclusion provides an effective method for the activation process and mechanism of brittle faults. Secondly, "fracture" occurs 41 times, "sandstone" occurs 40 times, "mechanism" occurs 33 times, and "permeability" occurs 32 times. They each have high frequency and low centrality, which also plays a very important role in this field. In coal mining, rock fractures often disturb the mining situation, and a local coal fracture will cause a lot of

deterioration in the coal. Therefore, mastering the fracture mechanical properties of coal plays an important role in studying the meso mechanism of this kind of disaster. For example, in 2019, Zhang, Y et al. [29] used a true triaxial compression test to study the evolution characteristics and mechanisms of the strain energy of hard rock. The state in a deep underground engineering environment will promote the understanding of the mechanisms of various geological disasters, hazards, and risks. Therefore, the keywords "fracture" and "mechanism" are the current hot topics. The research shows that in the project, the gas content at the bottom of the coal mine often exceeds the early warning value. The most fundamental reason for this is the penetration of formaldehyde gas. For example, Wen, Zhijie et al. [30] conducted a study in 2019 in order to accurately detect the development height of the water flow fracture zone (WFFZ) in the overburden, so as to ensure the safety and reliability of coal mining. Taking the coal seam under "Weishanhu in Jisan coal mine" as the experimental system, they studied the fracture law of the underwater overburden based on the flow stress damage model. Scholars began to combine acoustic emission technology with various experiments on coal damage. Therefore, "permeability" is a hot topic at present. The statistical effect of acoustic emission on fracture damage of coal materials has been studied in theory, experiment, and simulation. In recent years, based on acoustic emission, relevant research on the internal damage mechanism of materials has also been developed. At present, the research on the statistical effect of acoustic emission of different materials, their interactions, and their critical behavior should also be the focus of scholars. For example, Liu Hanlong et al. [31] studied the combination effect by using the combination of coal and sandstone, and also analyzed the acoustic emission energy of each layer and composite whole. Finally, it is concluded that compared with a single rock, the combined rock is more in line with the actual stratigraphic situation. The DISP series fully digital acoustic emission workstation is used to collect the acoustic emission signals of coal and sandstone layers in the samples under compression conditions. The probability density distribution of the signals is then analyzed.

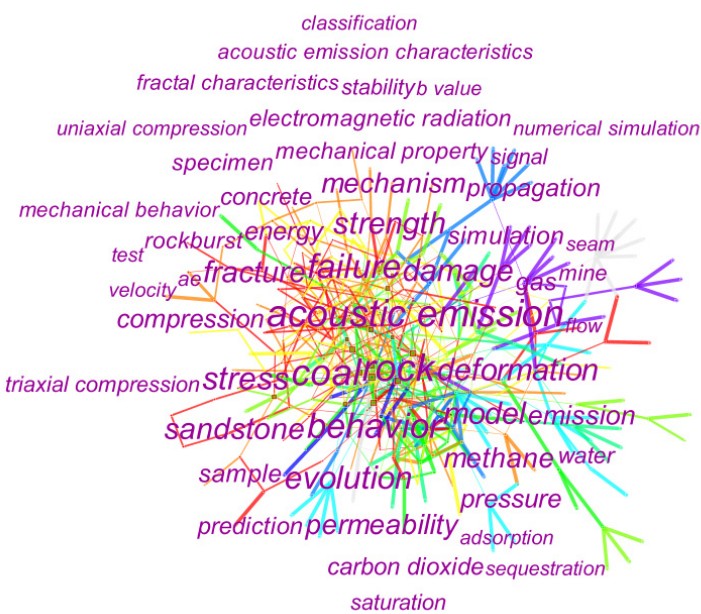

**Figure 9.** Keyword cooccurrence network.

### 3.3.2. Keywords Development and Change Analysis

Due to the strong relationship between keywords in various articles, using time to gather keywords over a period can intuitively obtain the trend of hot spots in this field. Figure 10 shows the change trend of keywords. Table 3 summarizes the occurrence frequency of keywords higher than 20 times in the table. The analysis results of Figure 10 and Table 3 are as follows:

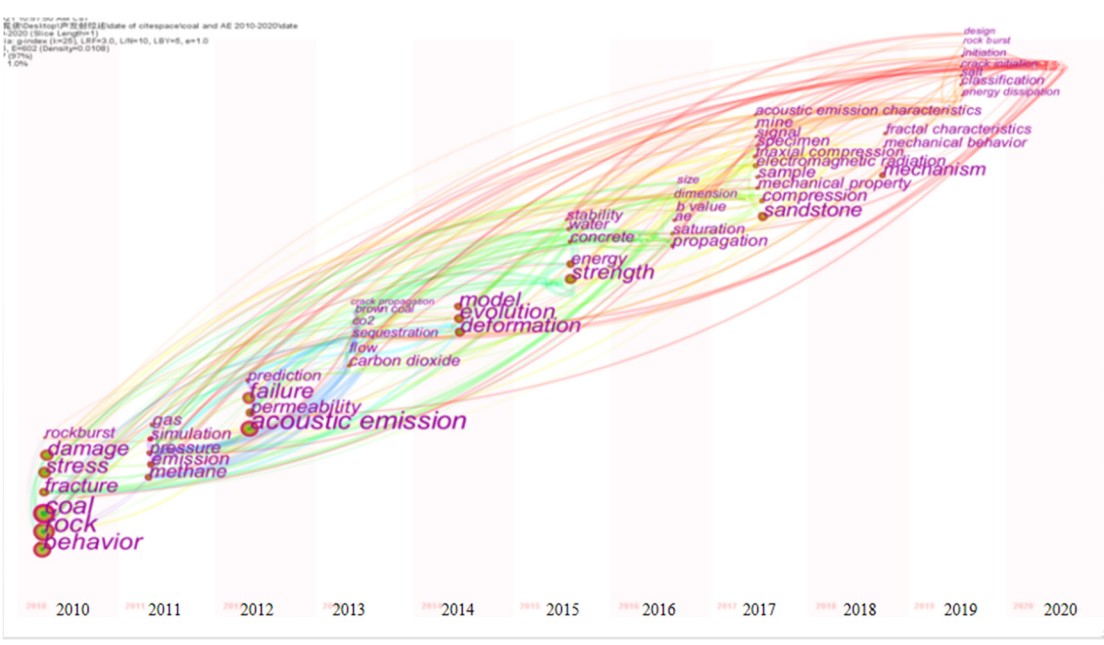

**Figure 10.** Time-zone view of the keyword cooccurrence network.

**Table 3.** Analysis of keyword three-stage change.

| 2010–2011 | Frequency | Centrality |
|---|---|---|
| rock | 125 | 0.15 |
| coal | 111 | 0.21 |
| behavior | 90 | 0.12 |
| stress | 52 | 0.08 |
| damage | 49 | 0.04 |
| fracture | 41 | 0.09 |
| emission | 26 | 0.15 |
| pressure | 22 | 0.16 |
| methane | 30 | 0.11 |
| simulation | 21 | 0.16 |
| **2012–2015** | | |
| acoustic emission | 121 | 0.11 |
| failure | 67 | 0.06 |
| permeability | 32 | 0.07 |
| model | 45 | 0.05 |
| evolution | 53 | 0.07 |
| deformation | 45 | 0.1 |
| strength | 46 | 0.06 |
| energy | 24 | 0.07 |
| **2016–2020** | | |
| sandstone | 40 | 0.05 |
| compression | 24 | 0.03 |
| mechanism | 33 | 0.03 |
| mechanical behavior | 12 | 0.02 |

From 2010 to 2011, the proportion of keywords "coal", "behavior", and "rock" was extremely high, with frequencies of 111, 90, and 125, respectively. Median centrality was 0.21, 0.12, and 0.15, respectively, with "coal" as the first, which was greater than 0.1. In Figure 10, since the size of the node is proportional to the frequency of documents where the key words are located, the lowest frequency of the node in the figure is 12. In this period, there were almost no occurrences of the keywords "acoustic emission," which indicates

that scholars mainly continue to use traditional methods when studying the properties of coal. During this period, scholars mainly studied the failure mechanism of coal, which laid a theoretical foundation for the later application of acoustic emission technology. For example, in 2010, Zhao Yixin et al. [32] analyzed the microstructure of coal samples by using a uniaxial cyclic loading test, thermal infrared radiation characteristics, X-ray CT, and scanning electron microscope SEM. Finally, they obtained the stress warning point easily in order to impact coal damage during the test. This is helpful to prevent damage to coal mines.

From 2012 to 2015, acoustic emission technology was accepted by people, and the application of acoustic emission in coal developed rapidly. The keywords "acoustic emission" began to appear in 2012, with a frequency of 121 times and a centrality of 0.11. Scholars have conducted relevant experiments and simulations through acoustic emission technology and applied it to practical engineering. For example, in 2012, Liu Yanbao et al. [33] studied the meso damage mechanism of aerated coal based on acoustic emission and established the constitutive model. Based on the Weibull theory and Mohr Coulomb law, the model fully considers the effective stress principle of solid gas coupling of aerated coal. The combination of a damage variable and a damage evolution equation with acoustic emission provides a clear physical engineering background. In 2013, Zhao, Yixin et al. [34] conducted an experiment and a numerical simulation of a fracture in coal under an impact load. A multi-spark high-speed photography system, an electronic microscope, a three-dimensional laser surface topography scanner, and X-ray microcomputer tomography were used to scan the coal fracture surface, and a discrete numerical simulation was carried out. Finally, it was concluded that the influence of heterogeneity and particle size in dynamic fracturing is more obvious than that of a quasi-static load. The keywords "permeability", and "model evolution energy" during this period show that scholars began to study the failure mechanism of coal from various angles, such as fracture energy in rock failure, water infiltration into the rock stratum, model establishment, and so on. For example, in 2013, Song, L et al. [35] used RFPA2D to simulate the dynamic load and other environments during coal mining. The stress field, acoustic emission, relative displacement, and seepage field around the fault are obtained respectively. The study found that without fault reinforcement, the river water easily flows into the goaf. It is suggested that a grouting method be used to strengthen the fault plane and surround the rock joints, which is of great significance to the suitability evaluation of dynamic infrastructure in goaf. In 2014, Cai, YD et al. [36] studied the change in permeability of coal under cyclic load, and found the change law of permeability through technology, which provided a theoretical basis for the effectiveness of coal degassing and $CO_2$ storage. In 2015, Lai, XP et al. [37] used FLAC (3D) to simulate the distribution and evolution of vertical displacement in giant rock columns. They analyzed the stability of rock mass structure, understood the structural dynamic model of giant rock columns, and provided an effective method for predicting disaster events related to structural instability.

The period from 2016 to 2020 was the mature stage of acoustic emission technology. Acoustic emission technology was not only applied to coal; scholars began to compare coal with other materials. For example, Chen, Lei et al. [38] used acoustic emission technology to study the compression deformation characteristics of siltstone under different water contents in 2017. Scholars discussed the relationship between the acoustic emission characteristics and the growth and expansion of siltstone fractures, and obtained the relationship between the water content of the specimen and the compressive strength and elastic modulus. They divided the deformation stages of the specimen and obtained certain rules. Chong, Zhaohui et al. [39] used numerical simulation to study acoustic emission events of argillaceous sandstone under confining pressure in 2017. They used the DEMAE model to calculate the scalar seismic tensor of particles in motion, and to further determine the magnitude of acoustic emission events. An algorithm for identifying the same space-time AE events is proposed, and the model is verified, which provides a research basis for

preventing earthquake disasters caused by coal mining. Various scholars have made great contributions to the research of acoustic emission in different periods.

### 3.3.3. Keywords Cluster Analysis

CiteSpace can provide users with clustering according to nouns or keywords in the literature. In this paper, keyword clustering is used to classify keywords according to the same topic, and divide the research data into different regional unit slices. Different regional unit slices represent the research hotspots in this field, and the size of different areas represents the number of articles. The clustering order starts with 0.

The main clusters with serial numbers 0 to 11 are selected in this study, as shown in Figure 11. They are the following: cluster #0 "acoustic aggregation", cluster #1 "$CO_2$ sequence", cluster #2 "propagation", cluster #3 "loading rate", cluster #4 "electromagnetic radiation", cluster #5 "saline aquifer", cluster #6 "strain energy", cluster #7 "wave", cluster #8 "hydro mechanical coupling", cluster #9 "aerosol nucleus", cluster #10 "noninterstructural short duration seismic signals". These 11 clusters contain most of the topics of the paper, and the regional unit slices of clusters are overlapped, indicating that these clusters are interrelated. It can be seen from the table that a parameter called silhouette score is located in the third column of the Table 4. Silhouette score is a parameter used to measure the quality of clustering performance, with a maximum of 1.00. It can be seen from Table 4 that the silhouette scores of the selected 11 clusters are not less than 0.80, indicating that these 11 clusters have a good matching degree with the main contents of the literature. The size in the second column represents the number of articles in the literature related to this cluster. The values for #0 "acoustic aggregation", #1 "$CO_2$ sequence", and #2 "propagation" are 44, 44, and 43, respectively, which occupy an important position in this study. Obviously, the three in this cluster are the hot topics in the current field of research. "Acoustic aggregation" represents the aggregation of sound waves. The basic principle of acoustic emission technology is to convert the elastic waves generated by acoustic emission sources into electrical signals continuously and in real time. Next, electronic equipment is used to display the required characteristic parameters, so as to know the internal defects of materials, which corresponds to "acoustic aggregation". Underground gas, coal dust explosion, and fire accidents are often encountered in the process of coal mining. The content of carbon dioxide in coal seams is high; sometimes coal and carbon dioxide outbursts occur. In a very short time, carbon dioxide is accompanied by a sudden and large amount of coal outbursts, which will produce a large amount of carbon dioxide, so #1 "$CO_2$ sequestration" is also a hot topic. Additionally, #2 "propagation" means the propagation of sound waves when cracks occur in coal. Compared with other rocks, the internal cracks of coal are larger, and too-large cracks will affect the propagation of sound waves. Scholars study algorithms to meet the needs of accuracy, which is also a hot topic in this field. In the relevant research, scholars mostly focus on the uniaxial compression and triaxial compression tests of coal, in which the loading speed is the main consideration, so #3 "loading rate" is also a pertinent issue at present. Finally, #4 "electromagnetic radiation" of coal is a phenomenon of outward radiation of energy. When coal deformation and failure under load occurs, coal will release strain energy. Therefore, acoustic emission and electromagnetic radiation have a certain relationship, so this is also a hot topic in the current field.

### 3.4. Cited Co-Occurrence Analysis

The cited analysis section is divided into three parts, as is the cooperative analysis. These are literature-cited co-occurrence analysis, author-cited co-occurrence analysis, and journal-cited co-occurrence analysis. The line in the figure represents that when an article quotes two articles at the same time, a connecting line will appear between the two articles. This means that the two articles have a reference co-occurrence relationship. The nodes in the reference network diagram represent the time of citation; the number of times cited corresponds to the size of the nodes.

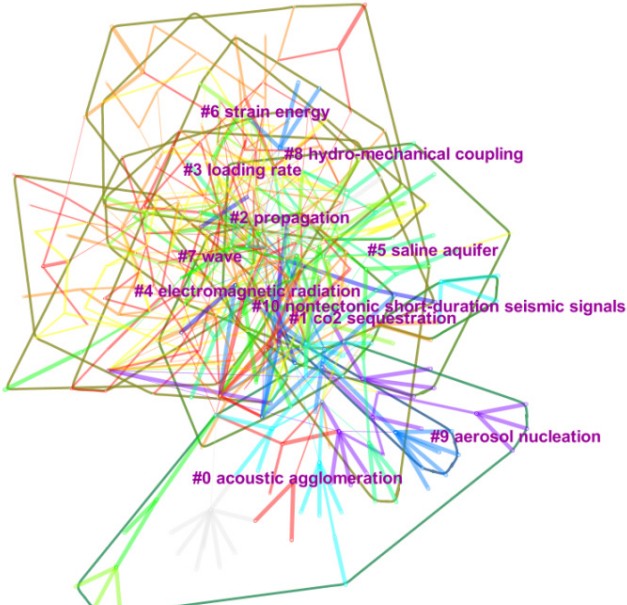

**Figure 11.** Main clusters in rockburst research of hard rock.

**Table 4.** Keyword clustering table.

| Cluster ID | Size | Silhouette Score | Cluster Label | Mean (Cite Year) |
|:---:|:---:|:---:|:---:|:---:|
| #0 | 44 | 0.914 | acoustic agglomeration | 2013 |
| #1 | 44 | 0.906 | co$_2$ sequestration | 2015 |
| #2 | 43 | 0.838 | propagation | 2014 |
| #3 | 36 | 0.802 | loading rate | 2017 |
| #4 | 32 | 0.880 | electromagnetic radiation | 2018 |
| #5 | 31 | 0.800 | saline aquifer | 2014 |
| #6 | 29 | 0.894 | strain energy | 2017 |
| #7 | 28 | 0.862 | wave | 2017 |
| #8 | 16 | 0.880 | hydro-mechanical coupling | 2017 |
| #9 | 8 | 0.984 | aerosol nucleation | 2012 |
| #10 | 6 | 0.987 | nontectonic short-duration seismic signals | 2015 |

### 3.4.1. Literature-Cited Co-Occurrence Analysis

Literature-cited co-occurrence analysis can be used to analyze influential papers in this field. Through literature-cited co-occurrence analysis, the key problems and important literature promoting the research in this field are known. CiteSpace is identified by the first author's name in the literature-cited co-occurrence analysis. Table 5 is the cited data table of articles, which can directly obtain the specific contents of the top ten articles in this field. According to Table 5, the cited articles of the top ten authors are mainly focused on experimental research. For example, the article shown in the table of Professor He, M.C., which is at the forefront, used acoustic emission technology to explore the rock core at 1140 m depth in the Jiahe Coal Mine under true triaxial experiment [21]. He summarized the dynamic damage process and characteristics of limestone from loading, to unloading, to failure. This directly explained the application of acoustic emission technology in the process of dynamic damage research, and provided a very effective tool for studying the characteristics of rockbursts in coal mines. Professor He, M.C. has many very important achievements in the research of large deformation of rock mass in mining engineering. He was the first person to reveal the whole process of rockbursts in the laboratory. He came to the conclusion that the rockburst strength is determined by the compressive elastic properties of high-stress rock mass, and the rockburst mode is controlled by the occurrence of a structural plane. He was elected as an academician of the Chinese Academy of Sciences

in 2013, and has made great achievements in this field. In Table 5, the team of Yang Sheng Qi studied the failure behavior of granite under temperature changes [40]. During the test, acoustic emission technology was used to monitor the evolution of internal cracks in the deformation process of granite, which provided a strong basis for scholarly research. Professor Yang Sheng Qi has been engaged in the basic research of rock mechanics and engineering application in the State Key Laboratory of deep geotechnical mechanics and underground engineering for many years, and has a great influence in the field of fractured rock mechanics. Cai Yidong et al. [36], in Table 5, combined with X-ray, acoustic emission, and ultrasonic technology, studied the permeability of coal under triaxial constraints. Because the permeability will affect the effectiveness of degassing before coal mining, the study of this problem is of great significance for coal mining. Acoustic emission technology is used to image the change of damage with applied stress during coal infiltration. Based on the complex reservoir geology of coalbed methane in China, Professor Cai Yidong focuses on the key scientific issues of coalbed methane enrichment and development. He has innovative knowledge in the research of coalbed methane reservoir microstructure, the gas liquid solid coupling mechanism, and the coalbed methane enrichment integration mechanism. These are very famous in this field. Generally speaking, the research on dynamic damage in the process of coal mining is the main research objective of scholars.

**Table 5.** Cited data analysis of the literature.

| Authors and Year | Title | Cited Frequency | Average per Year | Cited Frequency (2016–2019) | Cited Frequency in 2019 |
|---|---|---|---|---|---|
| He, M. C. et al. (2010) | Rockburst process of limestone and its acoustic emission characteristics under true-triaxial unloading conditions [21] | 451 | 37.58 | 302 | 74 |
| Yang, et al. (2017) | An experimental investigation on thermal damage and failure mechanical behavior of granite after exposure to different high temperature treatments [40] | 247 | 49.4 | 164 | 47 |
| Cai, YD et al. (2014) | Permeability evolution in fractured coalCombining triaxial confinement with X-ray computed tomography, acoustic emission, and ultrasonic techniques [36] | 134 | 16.75 | 103 | 17 |
| Kong, B. et al. (2018) | An experimental study for characterization the process of coal oxidation and spontaneous combustion by electromagnetic radiation technique [41] | 123 | 30.75 | 110 | 67 |
| Huang, BX et al. (2013) | The effect of loading rate on the behavior of samples composed of coal and rock [42] | 117 | 13 | 80 | 21 |
| Yan, FZ et al. (2015) | A novel ECBM extraction technology based on the integration of hydraulic slotting and hydraulic fracturing [43] | 116 | 16.57 | 90 | 24 |
| Vishal, V. et al. (2015) | An experimental investigation on the behaviour of coal under fluid saturation, using acoustic emission [44] | 115 | 16.43 | 90 | 27 |
| Kumari, W. G. P. et al. (2017) | Temperature-dependent mechanical behaviour of Australian Strathbogie granite with different cooling treatments [45] | 113 | 22.6 | 72 | 22 |

**Table 5.** *Cont.*

| Authors and Year | Title | Cited Frequency | Average per Year | Cited Frequency (2016–2019) | Cited Frequency in 2019 |
|---|---|---|---|---|---|
| Perera, MSAet al. (2013) | Effects of gaseous and super-critical carbon dioxide saturation on the mechanical properties of bituminous coal from the Southern Sydney Basin [46] | 96 | 10.67 | 70 | 20 |
| Perera, MSA et al. (2011) | Effects of saturation medium and pressure on strength parameters of Latrobe Valley brown coal: carbon dioxide, water, and nitrogen saturations [47] | 96 | 8.73 | 57 | 18 |
| Ranjith, PG et al. (2010) | The effect of CO2 saturation on mechanical properties of Australian black coal using acoustic emission [48] | 94 | 7.83 | 51 | 11 |

In addition to ranking the top ten articles cited from 2010 to 2020, the article also summarizes the literature cited in the past five years from 2016 to 2020. In Table 5, the number of citations of each article has changed, but enough remain to show that these articles play an important role in the research of acoustic emission technology, and the literature still occupies an important position at the forefront.

3.4.2. Author Cited Co-Occurrence Analysis

The data are imported into the CiteSpace software, and the reference mode is set to "cited journal." The node in the network diagram represents the author; the size of the node or font represents the number of times the author has been cited. The color of the circle represents the year of citation, which is similar to the outward arrangement of annual rings. The greater the annual thickness, the more times the author has been cited. The line in the network diagram represents that two authors are cited in the same article.

From the author's analysis of the network diagram (Figure 12), the attention of famous scholars in this field can be obtained. It can be found that the nodes of He MC, Kong XG, and Xie HP have the largest font and are cited the most times. Academician He MC is mainly engaged in the research of deep rock mass mechanics and engineering disaster control. He reproduced the entire process of rockburst for the first time in the laboratory. He came to the conclusion that the rockburst strength depends on the compressive elastic properties of high stress rock mass, and the rockburst mode is controlled by the occurrence of a structural plane. His has been evaluated by the academic community as a long-term in-depth study of the front line of mine production. He committed to the research and practice of the theory and the technology of large deformation disaster control. Professor Kong Xiangguo mainly studies coal and rock gas dynamic disasters, coal fracture, and geophysical signal response, which has a great impact on this field. Academician Xie Heping has long been committed to basic research and engineering practice in the fields of mining engineering, mine engineering mechanics, utilization of green energy, and deep earth science. In the field of coal, he established the mathematical model of top coal fragmentation control by using the fractal method and the energy dissipation theory. He proposed and designed a set of "top coal weakening pre-blasting technical schemes." This ensures the fragmentation requirements of top coal caving, improves the recovery rate, and makes the top coal caving technology successful under the condition of Datong hard coal. The citations of the three professors are 74, 70, and 66, respectively in Table 6. Professor He MC et al. [21] used acoustic emission technology to study the crack evolution process of gas coal as early as 2010. Professor Kong XG et al. [23] studied the critical slowing process of acoustic emission characteristics of methane-containing coal in 2015. It can also be seen in the table that other

professors have made good achievements in this field, and the research results of scholars play a great reference role in this field.

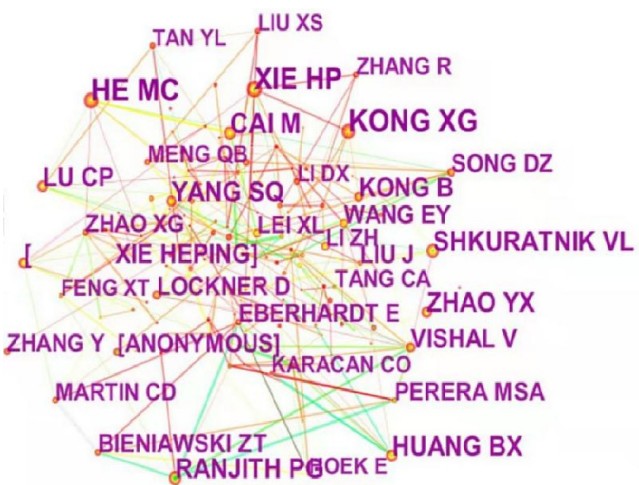

**Figure 12.** Author co-citation network.

**Table 6.** The dates of each cited author.

| Author | Cited Times | Year |
|---|---|---|
| He MC | 74 | 2012 |
| Kong XG | 70 | 2017 |
| Xie HP | 66 | 2016 |
| Cai M | 54 | 2010 |
| Yang SQ | 49 | 2013 |
| Ranjith PG | 47 | 2017 |
| Zhao YX | 46 | 2015 |
| Huang BX | 45 | 2015 |
| Shkuratnik VL | 44 | 2018 |

3.4.3. Journal Cited Co-Occurrence Analysis

It can be seen from the cited analysis network diagram of the journal in Figure 13 that INT J ROCK MECH MIN is located at the core of the network diagram, and has a very close relationship with other journals. Table 7 shows that it has been cited 343 times. Its full name is the "International Journal of Rock Mechanics and Mining Science". The impact factor in 2020 was 7.135 and the impact factor in the most recent five years was 7.042. It is the top journal in this field, followed by Rock Mech Rock Eng, Int J Coal Geol, Int J Min Sci Techno, Chinese Journal of Rock Mechanics And Engineering, Fuel, Eng Geol, and Journal of China Coal Society. These were cited 343 times, 264 times, 190 times, 172 times, 170 times, 165 times, 161 times, and 159 times, respectively. The above journals are well known at home and abroad. The influencing factors are 3.085, 1.767, 7.381, 7.042, 1.985, and 4.993. In addition, Table 7 shows that the influencing factors of the top three journals in the most recent five years are higher than 7. These three journals play a very important supporting role in the research on the application of acoustic emission technology. Secondly, the 453 articles published the top ten journals in the world are listed in Table 8 below. It is concluded that there is a certain connection between Tables 7 and 8. The "International Journal of Rock Mechanics and Mining Sciences" and "Rock Mechanics And Rock Engineering", which are cited at the top of the list, are also at the top of Table 7. Therefore, these two journals have made great contributions to the world.

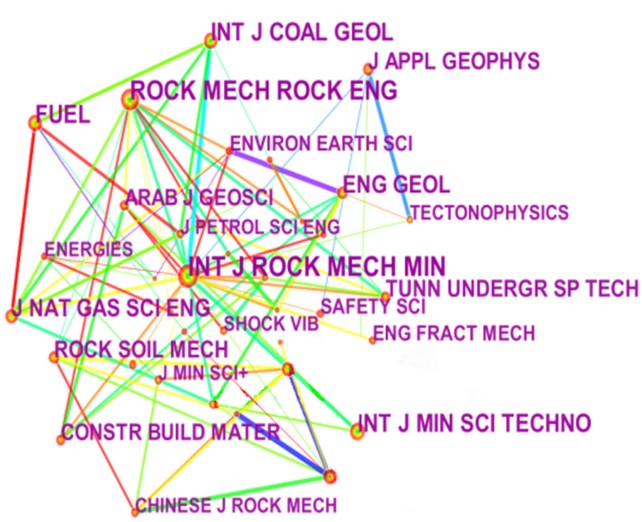

**Figure 13.** Journal co-citation network.

**Table 7.** Statistical table of the top ten cited journals.

| Journal Name | Country | Influencing Factors in Recent Five Years | Cited Times |
|---|---|---|---|
| International Journal of Rock Mechanics and Mining Sciences | Britain | 7.042 | 343 |
| Rock Mechanics and Rock Engineering | Austria | 7.381 | 264 |
| International Journal of Coal Geology | Netherlands | 7.173 | 190 |
| International Journal of Mining Science and Technology | China | 3.677 | 172 |
| Chinese Journal of Rock Mechanics and Engineering | China | 5.509 | 170 |
| Fuel | Britain | 6.63 | 165 |
| Engineering Geology | Netherlands | 7.138 | 161 |
| Journal of China Coal Society | China | 6.607 | 159 |
| Journal of Natural Gas Science and Engineering | Britain | 4.993 | 140 |
| Journal of Applied Geophysics | Netherlands | 2.264 | 116 |

**Table 8.** Statistical table of the number of articles published by the top ten journals.

| Journal Name | Country | Number | Influencing Factors in Recent Five Years | Percentage |
|---|---|---|---|---|
| Advances In Civil Engineering | Egypt | 25 | 1.923 | 5.519% |
| Energies | Switzerland | 25 | 3.085 | 5.519% |
| Shock And Vibration | U.S.A | 22 | 1.767 | 4.857% |
| Rock Mechanics And Rock Engineering | Austria | 21 | 7.381 | 4.636% |
| International Journal Of Rock Mechanics And Mining Sciences | Britain | 17 | 7.042 | 3.753% |
| Arabian Journal Of Geosciences | Saudi Arabia | 16 | 1.985 | 3.532% |
| Journal Of Natural Gas Science And Engineering | britain | 16 | 4.993 | 3.532% |
| Advances In Materials Science And Engineering | U.S.A | 13 | 1.94 | 2.870% |
| Journal Of Geophysics And Engineering | britain | 13 | 1.988 | 2.870% |
| Engineering Geology | Netherlands | 11 | 7.138 | 2.428% |

## 4. Sorting and Analysis of Network Diagram

The application of acoustic emission in coal from 2010 to 2020 is summarized as follows: Become familiar with the current research hotspots through keyword clustering, and summarize the top 11 hot terms in Figure 14. Through the keyword time zone map, according to the time division, become familiar with the entire development process of acoustic emission technology in coal, and divide the keywords for the development of acoustic emission technology in coal from 2010 to 2020 into three periods. It is known that the top 10 cited important papers have played a role in promoting the development of this

field. In addition, the research in this field, in the minds of scholars, is highlighted through the main institutions and major journals.

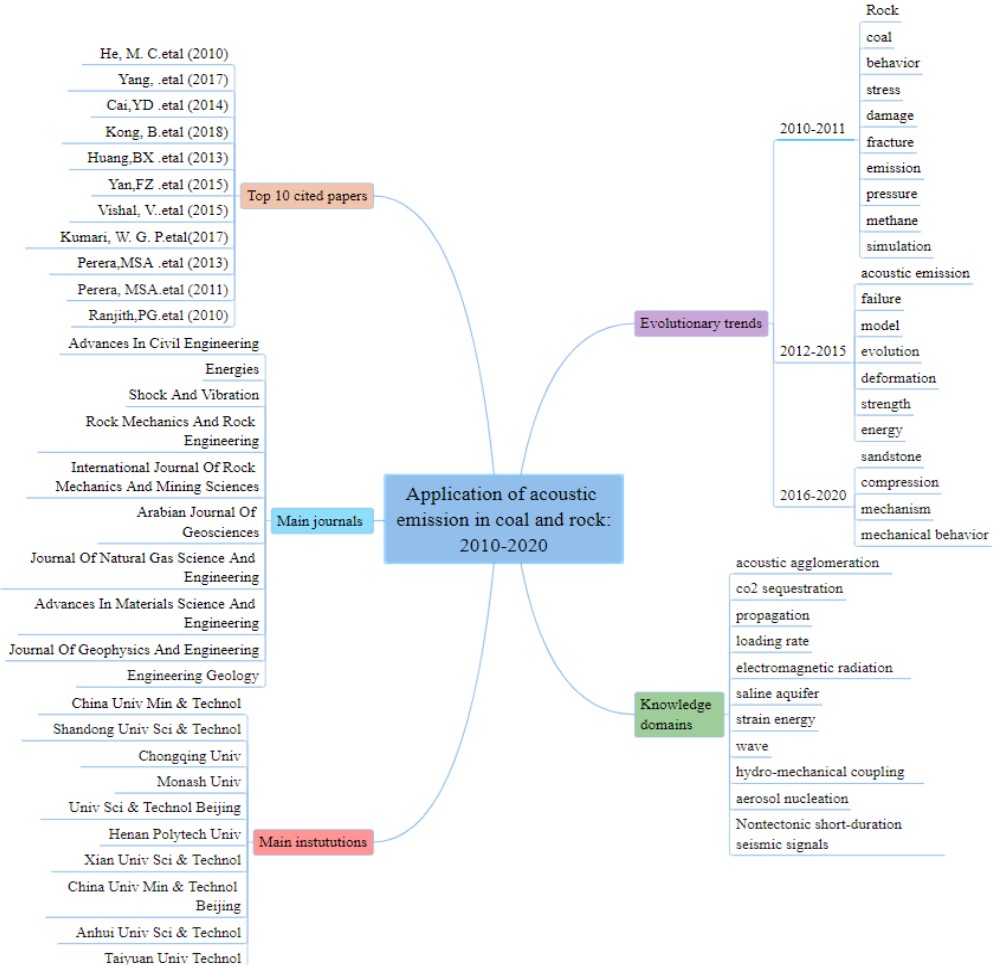

**Figure 14.** Summary chart of acoustic emission in coal: 2010-2020.

## 5. Future Research on Acoustic Emission in Coal Mining

The key research contents of acoustic emission technology in the coal field are divided into seven parts, which are shown in Figure 15. Based on the analysis, the future research on acoustic emission in coal mining is determined.

### 5.1. AE and Other Experimental Methods

By studying the articles on the application of acoustic emission in coal mining from 2019 to 2021, the main test methods of the 50 articles cited frequently are summarized in Table 9. It is known that the main experimental methods of acoustic emission in this field focus on uniaxial and triaxial tests. In triaxial tests, conventional triaxial tests are mainly used, with a small number of true triaxial tests. In the most recent three years, the number of uniaxial tests was higher than or equal to that of triaxial tests. The reason why the number of uniaxial and triaxial tests were equal in 2021 is that there were only four articles with relatively high citation times in 2021. In 2021, Liu, Shumin et al. [49] studied the influence of temperature shock on the mechanical properties of coal under a freeze–thaw test. It is combined with the principles of the uniaxial and triaxial compression tests. The mechanism analysis is carried out from a microscopic point of view in order to study the change rule of the mechanical properties of coal with temperature. The research results lay a theoretical foundation for studying the influence of liquid nitrogen on the mechanical properties of coal. For example, in 2021, Professor Hao Xianjie and others [50] used the crack volume and acoustic emission to analyze the anisotropy of crack damage, failure

mode, and acoustic emission characteristics under the uniaxial compression test. It was concluded that the cumulative energy method is more suitable for the acoustic emission cumulative counting method. This method can be used to determine the crack initiation strength and the damage strength of a coal seam. The article has been cited 25 times, ranking first in 2021. In 2021, Professor Zhang, R et al. [51] simulated the true triaxial stress state of on-site coal, and used acoustic emission technology to study the relationship between coal stress and gas pressure. The change rule and fractal characteristics of acoustic emission under different pressures and confining pressures are studied, and, finally, the fractal characteristics of acoustic emission time series are obtained, which can accurately reflect the damage evolution process of coal samples.

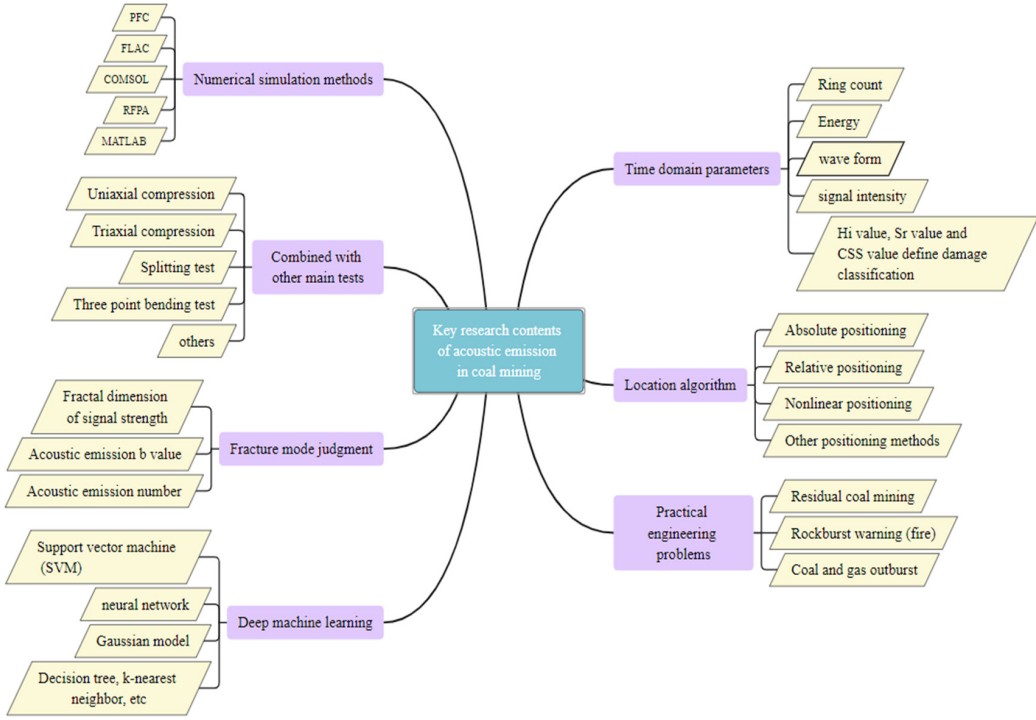

**Figure 15.** Key research contents of acoustic emission technology in the coal field.

**Table 9.** Occurrence times of acoustic emission and other main test methods in the most recent three years.

| Year | Uniaxial (Times) | Triaxial (Including True Triaxial and Conventional/Times) | Others (Three-Point Bending Test, Splitting Test, CT Scanning, Electromagnetic Radiation, etc./Time) |
|------|------------------|------------------------------------------------------------|-----------------------------------------------------------------------------------------------------|
| 2019 | 13 | 2 | 10 |
| 2020 | 13 | 6 | 11 |
| 2021 | 3 | 3 | 1 |

Secondly, the occurrence times of the three-point bending test and the splitting test are very few, and could only be obtained through retrieval in 2019. Professor Ma Dan studied the failure mode of roof in the process of underground mining [52]. This article was the most cited article in 2019, with 67 citations. The experimental method used was the point bending experiment. The crack evolution in the damage process is analyzed by ring counting in acoustic emission technology. Guo, Wei Yao et al. [53] studied the mechanical properties of specimens with different crack geometries in 2019, in combination with uniaxial compression and the Brazil test. They also used digital speckle correlation method (DSCM) and acoustic emission (AE) technology to record the deformation and failure process, and proposed the failure mechanism of the layered crack structure.

In addition, CT scanning and electromagnetic radiation technology are combined with the compression experiment to verify the phenomenon in many cases, and the number of separate occurrences is almost zero. For example, in 2021, Jiang, CB et al. used CT and acoustic emission to study the fracture evolution process of coal and shale under triaxial stress [54]. They divided the crack change of coal and shale under triaxial stress into stages, and, based on Rickman formula, they established a formula for calculating the modified brittleness index of test shale. In 2019, Chen, Shaojie et al. [55] carried out uniaxial compression experiment on coal shale combination and monitored the damage process of the sample through acoustic emission. They established an MFI model, and then analyzed the crack evolution process of the sample. For example, Professor Kong Xiangguo and Professor Wang Enyuan [56] studied the triaxial compression annual shrinkage failure of gas-bearing coal by using the fractal principle and acoustic emission technology in 2019. They used the acoustic emission counting principle to study the mechanical properties and dynamic damage mechanism of coal samples under a compression experiment. In 2019, Professor Wang Enyuan studied the acoustic emission characteristics under the effect of coal seam hydraulic flushing [57]. This study analyzed the gas flow and coal stress in the flushing process, so as to obtain the flushing effect.

The reason for the above situation is that uniaxial compression and triaxial compression mainly measure the compression of coal in the surrounding rock environment. In the splitting experiment and the three-point bending experiment, the indirect tensile strength and fracture properties of coal are mainly tested. The three-point bending often studies the strength change of impact-prone coal, as well as the fracture under overburden failure. However, limited to the universality of three-point bending equipment, the compression test is more suitable for the needs of practical engineering. In order to meet the needs of the actual project, the main experiments focus on uniaxial and triaxial. It is understood that the coal in the project is often in the situation of unequal triaxial stress, and the true triaxial test can better meet the requirements of the actual project. CT scanning, electromagnetic radiation, and other functions are relatively independent. Generally, they cooperate with the main test and play an auxiliary role in the test phenomenon. They are used alone for less time. The above analysis shows that in future experiments, uniaxial compression and triaxial compression will continue to be the main methods of analyzing the mechanical properties of coal under different conditions.

*5.2. Time Domain Parameters*

5.2.1. Acoustic Emission Ring Count

From Figure 15, we also know that in the experiment, the time domain parameters generated by acoustic emission are mainly used to monitor the rupture. The frequency domain waveform is used to quantitatively characterize the rupture, in which the time domain parameters include ringing count, energy, signal strength, HI value, Sr value, and CSS to define the damage type. Among them, ringing counts are widely used. For example, in 2019, Kong, B; Wang, EY et al. [56] qualitatively explained the damage evolution process of coal under load by using AE ring counting, which is helpful for explaining the damage mechanism of coal samples. AE ringing times can reflect the deflection and crack width produced by the sample loading process. Acoustic emission ring counting, combined with a height camera and an infrared thermal imager, can reflect the damage. The research results provide a theoretical reference for engineering disaster problems and prevention. The ringing times of acoustic emission can combine with other characteristics of acoustic emission, analyze the damage process of marble at different temperatures, reflect the development law of cracks in marble, and reveal the mechanism of rock thermal damage. In 2014, based on the triaxial compression test, Yang Yongjie et al. [58] studied the variation law of rock acoustic emission ring count under the state of surrounding rock. They pointed out that the research under the action of surrounding rock has more practical guiding significance. In addition, ring counting is also used to explain the difference between indoor and outdoor tests.

5.2.2. Energy and Waveform under Acoustic Emission Signal Intensity

The principle of acoustic emission technology is to monitor the elastic deformation energy of materials in deformation and failure. In a practical operation, the acoustic emission energy rate is often used to reflect the failure and crack propagation of coal. For example, Liu, SM et al. [59] studied the waveform, energy, and waveform characteristics of the acoustic emission signal received by coal fracture under the uniaxial compression test. Finally, the changing rule of acoustic emission waveform characteristics with the coal failure process is obtained, and the acoustic emission waveform characteristics can be used to evaluate the coal fracture process and shape. Kong XG, Wang EY et al. [56] reported the relationship between coal stress change and acoustic emission ringing count through experiments. This is a method often used by acoustic emission equipment to evaluate the rupture process of media. In addition, the acoustic emission signal is uncertain and sensitive to noise. The biggest problem with acoustic emission signal technology is the diversity of AE sources and the uncertainty of signals. Facing these problems, in 2018, Li Tianshu et al. [60] identified the intensity of the acoustic emission signal by using the characteristics of time series, and this quantitative prediction method can accurately reflect the change law of the research object. In this paper, the frequency domain feature analysis method is proposed, which can accurately describe the signal characteristics and different types of signals, and establish a network model for parameter analysis. It is pointed out that the final result is more accurate than other traditional machine learning signal classifications. In 2021, Cheng Tiedong et al. [61] proposed a new denoising method combining variational modal decomposition (VMD) and sample entropy (SE). The final test results can effectively remove the noise in the acoustic emission signal and help to identify the time domain characteristics of the signal.

In addition, HI value, Sr value, and CSS can define the damage type. These methods are widely used in concrete, but less used in coal. The reason is related to the properties of the materials, and the pores in coal are too large, so they are not suitable for these methods.

*5.3. Location Operation*

This mainly includes absolute positioning operation, relative positioning operation, nonlinear positioning operation, and other positioning operations. In acoustic emission, the specific location of coal damage is determined through the above positioning. Each location includes different calculation methods. Compared with other rocks, there are a large number of cracks in coal. More cracks will affect the propagation of the practical acoustic emission wave, and then affect the generation and expansion of micro-cracks in brittle materials under the action of external force. This has been a problem until now, without a more mature and accurate algorithm. As early as 2004, Li Guanghai of Tsinghua University and others conducted the positioning based on waveform analysis, from the theoretical point of view [62]. It has been proven that the group velocity measured in the experiment can realize more accurate positioning of the acoustic emission source. In addition, in the article on the application of acoustic emission in various materials by Álvaro Carrasco et al. [63], it is said that one of the more difficult problems in acoustic emission is the generation of a high volume of data on acoustic emission signals, which depend on the fixed threshold method for correct burst detection.

In 2021, the scholars [64] calculated the fracture of coal samples by using the Brune model and grid search method, then deduced the focal mechanism of sandstone and coal sample fractures based on the inversion of moment and source parameters. The final research results can be used to quantify and evaluate the focal characteristics of small-scale samples. Xiaoran Wang et al. [65] used the simplex method to locate the acoustic emission source in three dimensions. Since the AE event was physically interpreted as displacement discontinuity, the microcrack was expressed as a moment tensor. Both the quantitative moment tensor and the qualitative polarity analysis show the mixed mode mechanism of the microcrack, which represents the crack propagation of coal with local shear and tensile displacement. It is concluded that the acoustic emission energy distribution histogram is

helpful for studying the size fracture process zone of the object. Zhibo Zhang et al. [66] used the acoustic emission (AE) test system to locate the AE events of coal samples under an experimental uniaxial load. According to the b value and the basic theory of single chain cluster (SLC) method, the variation trend of relevant parameters with stress was studied.

However, these methods are not the best choice for crack monitoring. Therefore, developing a more correct burst detection method is a key step. Similarly, this theoretical problem is also common in coal mining, and should be studied in the future.

### 5.4. Acoustic Emission Fracture Style Judgment

Acoustic emission technology distinguishes the precursory style of coal fracture through different parameters. Among them, the fractal dimension, b value, and acoustic emission number of acoustic emission are widely used. It has been found that the waveform of coal has fractal law in the process of failure. The fractal model of acoustic emission intensity in the process of rock failure is established by using acoustic emission, which can provide a theoretical basis for the stability of rock mass before rock failure, and provide corresponding treatment means. For example, in 2019, Sun, H et al. [67] used the fractal characteristics of acoustic emission intensity to evaluate the damage degree between coal under dynamic load through the Grassberger Procaccia (G-P) algorithm. Finally, they proposed to apply the data mining method of correlation dimension to practical engineering. Zhao, Kang et al. [68] used the amplitude fractal characteristics of acoustic emission to monitor the failure of specimens. It has been found that in the process of large-scale cracks, the acoustic emission signals gradually tend to be orderly, and the fractal law is used to obtain the reference value. Research conducted by Dong Longjun et al. [69] shows that the change in b value is one of the important precursors of rock fracture. At present, with the maximum likelihood method, it is relatively simple to calculate the b value of rock acoustic emission, and the stability error is small. However, when the number of samples is small, the maximum likelihood method is not very stable, so it is important to focus on studying more accurate algorithms in the future. The acoustic emission number can obtain the failure style of coal, and is widely used. The cumulative acoustic emission number can reflect the failure mode of sandstone under splitting condition and compression condition. In 2011, Zuo Jianping et al. [70] conducted an acoustic emission test under a compression test by using rock, coal, and a coal combination. They proposed the number of acoustic emissions in a given time period. The number of acoustic emissions was segmented according to the stress increment, which can reflect the temporal and spatial evolution law of acoustic emission. It can also be used as a characteristic parameter to distinguish the failure mode of coal, which has a great influence in academia. Figure 16 and Table 10 show the stress-strain curve, acoustic emission, and the temporal and spatial evolution laws of coal assemblage in the article, which can clearly reflect the relationship between the stress-strain curve and the acoustic emission number.

### 5.5. Deep Learning of Acoustic Emission Machine

At present, a small number of scholars have conducted in-depth learning on acoustic emission machines, such as clustering, establishing Gaussian models, drawing neural networks, and obtaining more accurate values through some new algorithms. In 2020, Zhang, JF et al. [71] proposed an integrated machine learning method to accurately classify rockburst intensity. They changed the single index of rockburst prediction, such as back-propagation neural network, support vector machine, decision tree, k-nearest neighbor, logistic regression multiple linear regression, and naive Bayes, into a multi-index evaluation method which can meet the accuracy of practical needs and solve the classification of other problems in underground engineering. In 2021, Zhang Ruizhe et al. [72] used the compression test of yellow sandstone to study the parameters of the acoustic emission quiet period. They used this special period to explore the rock failure theory, in which the time domain data and frequency domain data of acoustic emission were regarded as the feature vectors, through the accuracy evaluation and model establishment analysis results.

Finally, the acoustic emission quiet period identification system was developed. In 2020, Xue Qingfeng et al. [73] studied an automatic classification method of acoustic emission data. The recognition accuracy of data in training reached 100%. This method establishes a neural network suitable for rock tests, which are convenient for automatic classification of different types of data. In the future, this will become the focus of in-depth research by scholars.

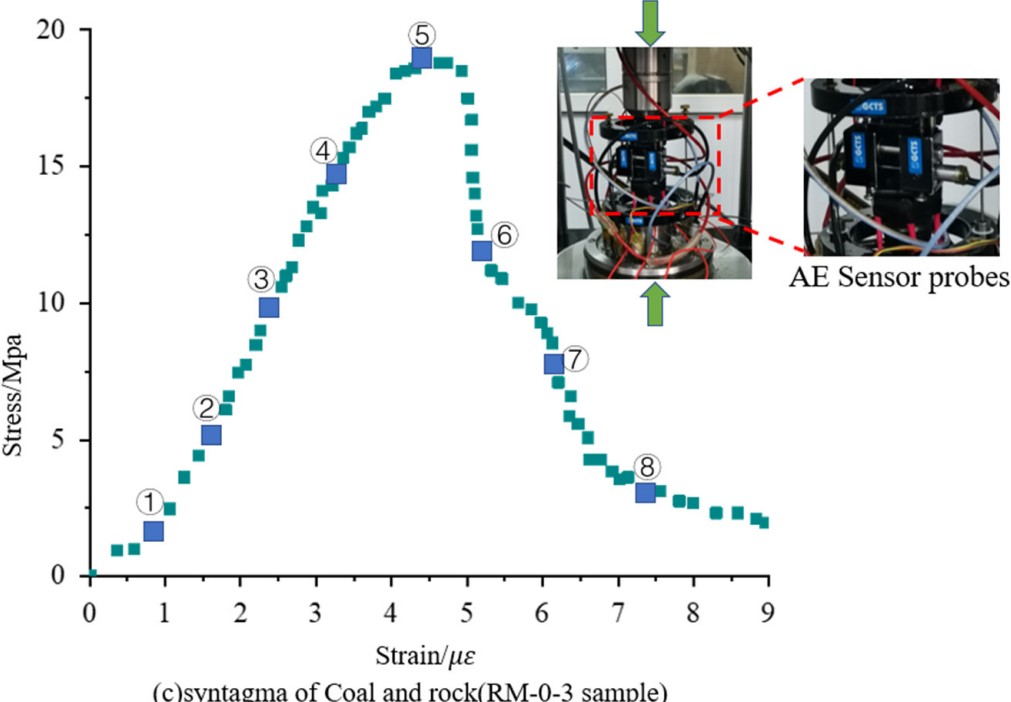

(c)syntagma of Coal and rock(RM-0-3 sample)

**Figure 16.** Stress-strain curve and space-time evolution law of acoustic emission [70].

**Table 10.** Evolution law of acoustic emission at different loading stages.

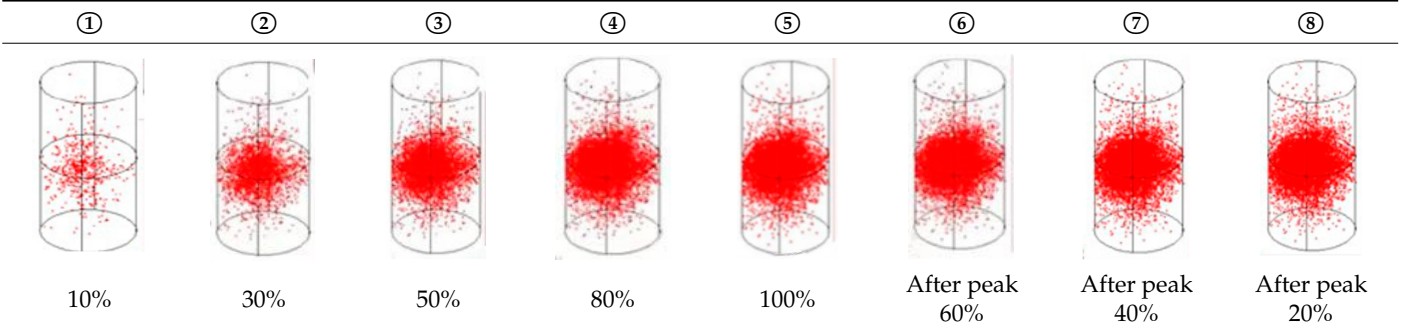

| ① | ② | ③ | ④ | ⑤ | ⑥ | ⑦ | ⑧ |
|---|---|---|---|---|---|---|---|
| 10% | 30% | 50% | 80% | 100% | After peak 60% | After peak 40% | After peak 20% |

## 5.6. Practical Engineering Problems

Acoustic emission technology in coal mining mainly aims at the problems of water inrush early warning, rockburst early warning, and coal and gas outburst. As early as 2003, Yao Jianguo et al. [74] said that the conditions of high water pressure, high stress, high gas, and high temperature in deep mining should be taken as the key research object. They mentioned the Subsidence Reduction mining technology. In 2021, Professor Zhao Yangsheng reviewed the development of rock mechanics, which described some unsolved problems [75]. The author lists the development of underground mining from the 17th century to the present. He points out that the achievements of coal mining are basically aimed at the mining of coal seams with near horizontal or slight angles. There are very few

deformation laws for coal seams with very large dip angles and surrounding rock, which is also the fundamental reason why steep coal seams have not been mined for a long time.

In terms of residual coal mining, Feng Guorui et al. [76] proposed in 2020 that great progress has been made in the research of residual coal mining. However, many complex problems will still be faced in the mining. For example, in the process of coal mining, due to the mutual connection of water source, channel, and ponding volume, the hydrogeological situation is extremely complex, leading to issues such as the problem of ponding and gas accumulation in the old goaf. In view of these problems, the detection technology of coal and hazard source technology need to be solved.

In terms of coal and gas outbursts, according to the research, in the project, the gas content at the bottom of the coal mine often exceeds the early warning value. Although the current risk has been greatly reduced, the gas with high concentration often appears in the goaf behind the working face. It actually poses a threat to the safety of workers. In this regard, it still needs to be strengthened. Cao Kang et al. [77] put forward the problem of coal and gas outbursts in 2020. Researchers put forward many corresponding treatment methods, but the accuracy of the data still needs to be improved. They determined the intensity of acoustic emission through the outburst to be dangerous in the event of "11227 transportation roadway" in the Jinjia coal mine. The comprehensive gas early warning method, between electromagnetic radiation intensity, is used to monitor the working face. Finally, they obtained a high accuracy rate of 92.7%, which is of great significance to the actual production.

Rockburst has always been a typical problem in coal mining. The existing achievements in rockburst include quantitative prediction of coal pillar strain rockburst, quantitative prediction of surrounding rock strain rockburst, and prediction of fracture slip rockburst. In order to solve the problem of rockburst, He Man Chao et al. [21] developed a new type of constant resistance and large deformation bolt with Poisson's ratio effect. However, there are many problems, including the complex environment, in actual mining. It has been pointed out that at this stage, the prediction of rockburst by acoustic emission technology is still in the comparison stage between post-disaster data and monitoring data. The technology is not mature; it is in the qualitative understanding stage, and the intelligent monitoring and early warning technology needs to be developed.

*5.7. Numerical Simulation Methods*

Due to the heterogeneity, nonlinearity, and discontinuity of coal, in-situ tests are usually needed to obtain the basic mechanical parameters of coal. However, due to the complexity of practical engineering and the high cost of experimental research, especially the poor repeatability of test results, only through the in-situ test can the geological information and the real law of rock mass mechanics in rock mass engineering be reflected. Numerical simulation has a wide application prospect in rock mass mechanics because of its controllability, non-destructive properties, safety, and allowance of multiple repetitions. Moreover, scholars at home and abroad have achieved many research results in this regard. Common numerical simulation methods include FLAC, PFC, COMSOL, RFPA, and MATLAB. The scholars used PFC2D to evaluate the cracking behavior of intact coal samples under biaxial compression, and the numerical results show that there are cracks in coal, which can be divided into six modes: shear, tension, secondary shear crack, secondary tension crack, shear-dominated mixed crack, and tension-dominated mixed crack [78]. Feng Du et al. [79] established the internal mineral distribution network and cracks of coal by using FLAC3D modeling under a uniaxial test. The model was verified by fractal geometry, and the damage equation of coal was established under a compression test by using FLAC3D. Some scholars studied the strength distribution and failure characteristics of coal under uniaxial compression by using COMSOL software, and realized the heterogeneity and destruction of coal by using the MATLAB numerical method [80]. Others used RFPA to simulate the rock failure process under an external load. They simulated rock acoustic emission (AE) and provided certain guidance for structural design and different working conditions [81].

In 2018, some scholars pointed out in the literature that for isotropic porous media, the one-dimensional seepage control equation with threshold pressure gradient (TPG) in the Cartesian coordinate system and the cylindrical coordinate system is incompatible [82]. It was suggested that the equations composed of pressure and seepage velocity vectors should be used in the numerical simulation of anisotropic porous media with TPG.

Scholars use numerical simulation to provide reference data for coal mining, so as to be applied to practical engineering. In the future, scholars will still have a deeper exploration on the application of numerical simulation.

## 6. Conclusions

Through four modules of cooperative analysis, keyword co-occurrence, co-citation analysis, and future research of acoustic emission in coal mining, the conclusions are as follows:

(1) In the part focusing on cooperation analysis, it is obvious that China plays a very important role in the whole field, and has links with other countries. The proportion of documents published by Chinese scholars is as high as 90%. Among them, 168 articles were issued, led by China University of Mining and Technology. The author, led by Professor Yuan Wang, has 40 articles.

(2) In the part of keyword co-occurrence and trend analysis, "rock" and "acoustic emission" occupy the main positions in the keyword co-occurrence network diagram. During the stage from 2012 to 2015, acoustic emission technology was accepted by people, and the application of acoustic emission in coal developed rapidly. From 2016 to 2020 was the mature process of acoustic emission technology, and acoustic emission has been applied to all aspects of engineering.

(3) With regard to the cluster analysis, the number of #0 "acoustic aggregation", #1 "$CO_2$ sequestration", and #2 "propagation" documents in the keyword clustering graph is 44, 44, and 43, respectively, which corresponds to the research topic and is a pertinent issue in this field.

(4) In the citation analysis of the third part, it was learned that Professor He, MC's "rockburst process of limestone and its acoustic emission characteristics under true triaxial unloading conditions" is the most cited article, with 451 citations. The corresponding author, He, MC, is the most cited author. The "International Journal of Rock Mechanics and Mining Sciences" is the most cited Journal.

(5) In the future research of acoustic emission in coal mining:

Uniaxial and triaxial compression tests will still be the main test methods; and the principle problem of acoustic emission location operation will become the focus in the future.

At present, the maximum likelihood method for calculating b value is not very stable, so we should focus on studying more accurate algorithms to solve this problem in the future.

In practical engineering problems, there is little research on the deformation activity law of steep coal seams and surrounding rock. For practical engineering, the existing technology is still immature; intellectualization and accuracy based on experience will allow great room for improvement in the future.

In addition, the above analysis is only based on the data retrieved by the Web of Science system. In the future, a more comprehensive knowledge network diagram can be combined with other types of databases. In addition, keywords with the same meaning in CiteSpace software should be considered for classification and evolution.

**Author Contributions:** All authors conceived the research idea and the framework of this study. Q.C. prepared the first draft, J.Z. edited and modified the article, L.Y. analyzed the structure of the article, S.Z. and K.Q. collected and sorted out papers related to the theme of the article. All authors have read and agreed to the published version of the manuscript.

**Funding:** This work was supported by National Natural Science Foundation of China Grants no.: 41877257, Beijing Outstanding Young Scientist Program (BJJWZYJH01201911413037), China Coal Technology Engineering Group Co.,Ltd Technology Innovation and Entrepreneurship Fund Special Pro-ject(2020-2-TD CXY005), and the Fundamental Research Funds for the Central Universities(2022YJSLJ02), and State Key Laboratory of Strata Intelligent Control and Green Mining Co-founded by Shandong Province and the Ministry of Science and Technology, Open Fund Pro-ject, Research on Mutual Disturbance Erosion Mechanism and Prevention and Control Technology of Inter-well Joint Mining under Deep Complex Geological Conditions, Project Approval No. SICGM202106.

**Institutional Review Board Statement:** Not applicable.

**Informed Consent Statement:** Not applicable.

**Data Availability Statement:** Not applicable.

**Conflicts of Interest:** The authors declare no conflict of interest.

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
