# Peer review of "A Review on Application of Acoustic Emission in Coal—Analysis Based on CiteSpace Knowledge Network"

_processes, doi:10.3390/pr10112397_

Round 1
Reviewer 1 Report
Dear Authors,
I think that it is valuable work. Some small corrections are necessary to be provided:
1) what das mean lack of continuity after "and" in line 4? One author has been missed?
2) WoS - Web of Science should be written in original, correct form - see line 16 in abstract.
3) Mistakes in the names of institutions appears in further parts of the article - Figure 1, p.4.
Please correct this.
Author Response
Thank you for your suggestions. We have explained and corrected the questions. Please refer to the attachment.

Reviewer 2 Report
Please see these comments for the manuscript.
1. The language of the manuscript needs a comprehensive modification.
2. Line 35-40: It seems to be repeating the same information. Consider revising it.
3. No need to mention coal rock throughout the texts. Simply coal is good enough.
4. Section 1.1: Revise the language of the section. It does not represent any meaningful information. What do the authors mean by “disasters between coal and gas”? In line 74, consider revising the phrase simulation experiment.
5. Line 78: Acoustic emission number means AE counts/Hits?
6. Section 1.2: Mechanical mechanism?
7. Line 183: Looks like wrong information. The concept and application of AE to rocks were established long back. Not 2010.
8. A question arises on the reliability of the software. What if the software is not capable of gathering all the information? There is no information on the reliability of the software.
9. What is the applicability of section 3.1.2. Any new science or information?
This manuscript does not add any information or new sciences; instead, twisting and illustrating the pre-existing literature using software whose reliability is also under question.
Author Response
Dear reviewer:
Thank you very much for considering reviewing our manuscript. All authors would like to take this opportunity to express our gratitude to you for the valuable suggestions on our manuscript (Manuscript number: processes-1797071, Title: "A review on application of acoustic emission in coal --Analysis based on CiteSpace knowledge network"). Those comments are all valuable and helpful for revising and improving our manuscript, as well as the important guiding significance to our research. We have carefully studied comments and have made the correction which we hope to meet with approval. We mark all the changes in red in the revised manuscript. Here are the responses to the your comments.
We look forward to your decision on it as soon as possible.
Best regards.
Sincerely yours
Shankun Zhao, Qian Chao, Liu Yang, Kai Qin, Jianping Zuo*

Reviewer 3 Report
After reading the manuscript “A review on application of acoustic emission in coal --Analysis based on CiteSpace knowledge network”, my comments are below;
1. The authors must review the article for minor but important mistakes like 5th author name?
2. Why the authors have defined the time span. Justify it in the article.
3. The authors have discussed the fault slip type of rock burst, however, citations are not available. The reviewer recommends that some relevant citation must be included the following related article or some other similar article in the study.
4. What will be the contribution of this study in the field of rock engineering in general and coal mining in particular? Justify.
5. Show the justification for “CiteSpace software visualizes 453”.
6. The authors have only focused on the acoustic emission as the paper is reviewing the phenomenon. However, the author should also include other approaches used for predicting failure like energy evolution, infrared radiation, etc.
7. Keeping the title of the paper, the paper must include the application, limitation, future trend trends review in the article. Also, comparison of technology and methodologies.
8. Statistical analysis of figures 7-11 have no practical meaning. Instead of such analysis, the authors can show the major findings of the technologies and their practical application.
9. No role of Table 5 in rock engineering. The actual role is what major development have been devised, developed and applied.
10. As rock engineering researchers, the trends must be specific to the application of mechanic’s rock and rock mass under loading, unloading etc.
Author Response

(The authors gave the same response as above.)

Round 2
Reviewer 2 Report
Changes have improved the MS.
Reviewer 3 Report
1. Stick slip instability?
2. Fault slip type rock burst is not the only issue in mining projects. Underground civil excavation also has the issue. For reference, fault slip rock burst discussed in details in the following articles in civil projects;
· Static and dynamic influence of the shear zone on rockburst occurrence in the headrace tunnel of the Neelum Jhelum hydropower project, Pakistan
· Impact of Construction Method and Ground Composition on Headrace Tunnel Stability in the Neelum–Jhelum Hydroelectric Project: A Case Study Review from Pakistan
3. Minor grammatical and linguistic mistakes are available.
4. What will be the practical application of this study in rock engineering.
5. Discuss in details the major development that have been devised, developed and applied.
Author Response
Dear reviewer:
Thank you very much for considering reviewing our manuscript again. All authors would like to take this opportunity to express our gratitude to you for the valuable suggestions on our manuscript (Manuscript number: processes-1797071, Title: "A review on application of acoustic emission in coal --Analysis based on CiteSpace knowledge network"). Those comments are all valuable and helpful for revising and improving our manuscript, as well as the important guiding significance to our research. We have carefully studied comments and have made the correction which we hope to meet with approval. We revised it on the basis of the review report (round 1), and all the changes were marked in green in the revised manuscript. The following is a response to your comments.
We look forward to your decision on it as soon as possible.
Best regards.
Sincerely yours
Shankun Zhao, Qian Chao, Liu Yang, Kai Qin, Jianping Zuo*

Round 3
Reviewer 3 Report
Accept in current